# Computer-Aided Design as Language

**Yaroslav Ganin**[1]* **Sergey Bartunov**[1] **Yujia Li**[1] **Ethan Keller**[2] **Stefano Saliceti**[1]

[1]DeepMind     [2]Onshape

## Abstract

Computer-Aided Design (CAD) applications are used in manufacturing to model everything from coffee mugs to sports cars. These programs are complex and require years of training and experience to master. A component of all CAD models particularly difficult to make are the highly structured 2D sketches that lie at the heart of every 3D construction. In this work, we propose a machine learning model capable of automatically generating such sketches. Through this, we pave the way for developing intelligent tools that would help engineers create better designs with less effort. The core of our method is a combination of a general-purpose language modeling technique alongside an off-the-shelf data serialization protocol. Additionally, we explore several extensions allowing us to gain finer control over the generation process. We show that our approach has enough flexibility to accommodate the complexity of the domain and performs well for both unconditional synthesis and image-to-sketch translation.

## 1 Introduction

Computer-Aided Design (CAD) is used in the production of most manufactured objects: from cars to robots to stents to power plants. CAD has replaced pencil drawings with precise computer sketches, enabling unparalleled precision, flexibility, and speed. Despite these improvements the CAD engineer must still develop, relate and annotate all the minutiae of their designs with the same attention to detail as their drafting-table forebears. CAD productivity might be improved by the careful application of machine learning to automate predictable design tasks and free the engineer to focus on the bigger picture. The flexibility and power of deep learning is uniquely suited to the complexity of design.

Sketches are at the heart of mechanical CAD. They are the skeleton from which three dimensional forms are made. A sketch consists of various geometric *entities* (*e.g.*, lines, arcs, splines and circles) related by specific *constraints* such as tangency, perpendicularity and symmetry. Figure 1 illustrates how entities and constraints work in tandem to create well-defined shapes. Geometric entities lie on a single plane and together form enclosed regions used by subsequent construction operations such as lofts and extrusions to generate complex

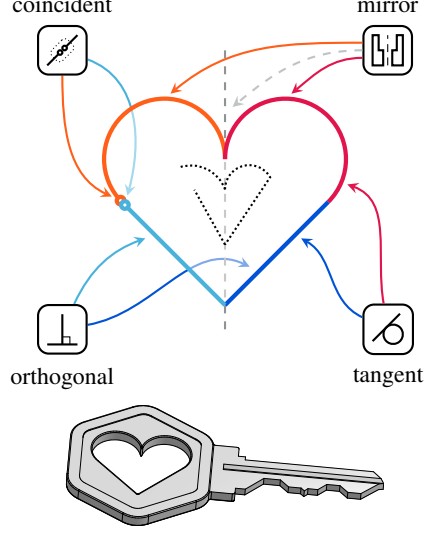

Figure 1: **The anatomy of a CAD sketch**. Sketches are the main building block of every 3D construction. A sketch consists of *entities* (*e.g.*, lines and arcs) and *constraints* (*e.g.*, tangent and mirror). The *dotted curve* shows what happens if we drop some of the constraints — the design idea is lost.

---

*Correspondence to: ganin@deepmind.com.

35th Conference on Neural Information Processing Systems (NeurIPS 2021).

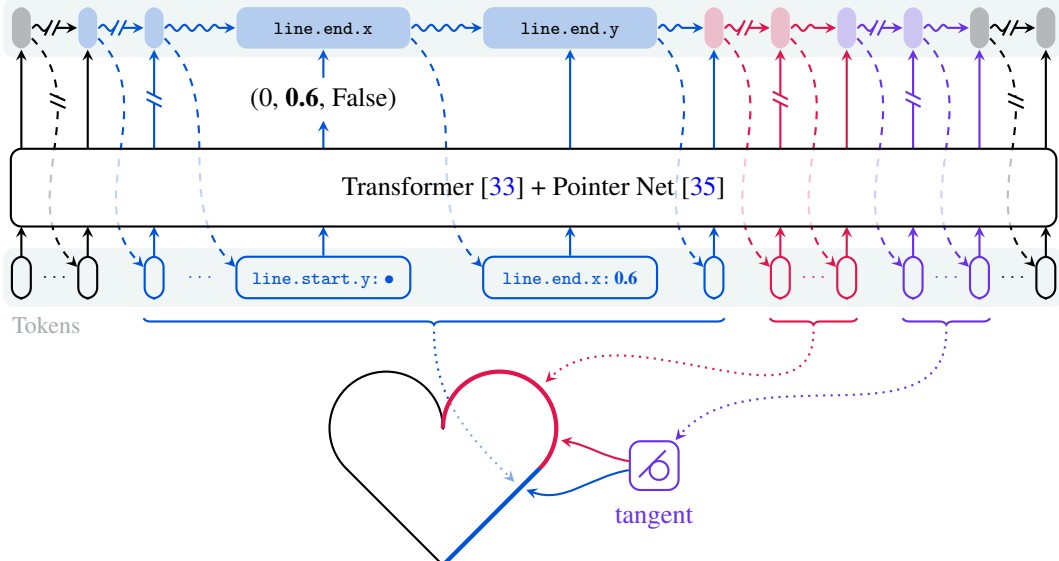

Figure 2: **Interpreter-guided generation of a sketch.** At each point in time, a Transformer [33] outputs a raw value which is fed into an *interpreter* that decides which field of a *Protocol Buffers* message this value corresponds to. Once the field is populated the interpreter communicates (- ➤) its decision back to the Transformer and transitions (⤳) to the next state.

3D geometry. Well-chosen sketch constraints are essential to properly convey design intent [2] and facilitate the sketch's resilience to successive parameters modifications which is often understood as a measure of the quality of a design document [8]. The dotted curve in Figure 1 shows what happens when some of the constraints are dropped – the design idea is lost.

The complexities of sketch construction are analogous to those of natural language modeling. Selecting the next constraint or entity in a sketch is like the generation of the next word in a sentence. In both contexts, the selection must function grammatically (form a consistent constraint system in the case of the sketch) and work towards some cohesive meaning (preserve design intent). Luckily, machine learning has proved highly successful in generating natural language — especially the Transformer [33] trained on vast amounts of real-world data [26, 6]. It is therefore a promising choice for adapting to the task of sketch generation. This work is our take at this adaptation.

We make the following contributions: **(1)** We devise a method for describing structured objects using Protocol Buffers [32] and demonstrate its flexibility on the domain of natural CAD sketches. **(2)** We propose several techniques for capturing distributions of objects represented as serialized Protocol Buffers. Our approach draws inspiration from recent advances in language modeling while focusing on eliminating data redundancy. **(3)** We collect a dataset of over 4.7M of carefully preprocessed parametric CAD sketches and use this dataset to validate the proposed generative models. To our knowledge, the experiments presented in this work significantly surpass the scale of those reported in the literature both in terms of the amount of training data and the model capacity.

## 2   Related work

**Datasets and generative models for CAD.**   Until recently there were very few parametric CAD datasets large and varied enough to serve as training data for machine learning. This situation had started to change with the release of the ABC dataset [16] containing a collection of 3D shapes from the Onshape public repository [22]. Unfortunately, the main focus of [16] revolves around meshes and, as a result, the dataset is difficult to use for sketch modeling.

Several works concurrent with ours deal with the symbolic representation of CAD constructions. Seff et al. [31] center their attention on contributing a better dataset of 2D sketches but also provide a proof-of-concept model predicting a selected subset of object attributes. Willis et al. [37] use the

dataset from [31] to train a modification of [21] which demonstrates a boost in generation quality but is designed to only work for sketch entities. The latter is addressed in our present work and in a subsequent paper by Para et al. [24]. While [24] employ very similar ideas to ours they do not support certain features of the CAD data and their proposed model is a more direct adaptation of [21] and therefore cannot handle arbitrary orderings of entities and constraints. In Section 5, we show that object ordering has a substantial impact on the performance.

Fusion 360 Gallery [36] attacks CAD data from a different angle. Here, the task is to recover a sequence of extrusion operations that gives rise to a particular target 3D shape. Despite dealing with 3D, this setting is deliberately limited: sketches are assumed to be given and the proposed model only decides on which sketch to extrude and to what extent. [38] considers a more general scenario where extrusion profiles are not provided and need to be synthesized from scratch. Although both of these works make initial steps towards full parametric CAD generation, they rely on significant simplifying assumptions and therefore it is unclear how well they will scale to more real-world scenarios. Our approach, on the other hand, is designed to be flexible and domain-agnostic and is only limited by the data availability.

**Vector image generation and inference.** Synthesizing CAD sketches bears a lot of similarities with predicting vector graphics. In this field, several recent works Carlier et al. [7] and Reddy et al. [30] using different vector object representations to define generative models of vector images. Egiazarian et al. [13] take a more traditional computer vision approach and propose a multi-stage pipeline for vectorizing technical drawings. All of these methods use highly domain-dependent architectures and, therefore, it would be a non-trivial task to adapt them for generation of complex sketch objects. CAD community has also been concerned with a similar task of image to CAD conversion [20, 34, 10, 11], largely focusing on heuristic object recognition while our work relies more on learning the recognition from data.

**Transformers for sequence modeling.** In our work, we employ Transformers [33] as a computational backbone for the proposed approach. Due to its scalability and excellent performance [29, 9, 6, 27], this architecture has become the dominating approach in many sequence modeling applications. Our method can be seen as generalization of PolyGen [21], a Transformer-based generative model for 3D meshes. Similarly to [21], we use Pointer Networks [35] to relate items in the synthesized sequence. Unlike PolyGen, however, our framework can handle non-homogeneous structures of arbitrary complexity. Moreover, we simplify the architecture to use a single neural network to generate the entire object of interest. All these improvements make our approach a good fit for modeling CAD sketches and potentially other components of CAD constructions.

## 3 Data

Formally, a CAD sketch is defined by two collections of objects: entities and constraints. Each object is generally represented as a set of attribute-value pairs where a value can be either primitive (*e.g.*, integer or floating-point) or complex (*e.g.*, an array or another object). Sketches that we use in this work originate from the Onshape platform [22] which provides them in JSON format [25]. As the first step in our processing pipeline we convert JSON messages into Protocol Buffers (PB) [32]. In order to keep the pipeline as domain-agnostic and as widely applicable as possible, we aim to avoid any significant changes to the data and largely retain the original structures of objects. The benefit of converting into PB is twofold: the resulting data occupies less space because unnecessary information is removed, but also, unlike JSON, PB provide a convenient way to define precise specifications for structures of arbitrary complexity.

Listing 1a shows how we represent the *line entity* and the *mirror constraint* (see Appendix A for an extensive list of supported objects). The line specification is straightforward: we first need to decide whether our entity should be treated as a construction geometry[2] and then provide pairs of coordinates for the beginning and end of the segment. The `MirrorConstraint` is used to force an arbitrary number of pairs of geometries (*i.e.*, `mirrored_pairs`) to be symmetrical with respect to some axis (*i.e.*, `mirror`). Constraints rely on the `Pointer` data type to specify entities they act upon. In practice, a pointer is simply an index in the table of all the eligible pointees (*i.e.*, entities and their parts).

---

[2]Construction geometries are rendered in dashed style in the figures.

```
message LineEntity {
  bool is_construction = 1;
  message Vector { // 2D coordinate.
    double x = 1;
    double y = 2;
  }
  Vector start = 2; // Start point.
  Vector end = 3; // End point.
}
```

```
message MirrorConstraint {
  Pointer mirror = 1; // Axis of symmetry.
  message Pair { // Mirrored objects.
    Pointer first = 1;
    Pointer second = 2;
  }
  repeated Pair mirrored_pairs = 2;
}
```

(a) **Entities and constraints** have similar structures. `Pointers` refer to entities that constraints are applied to.

```
message Entity {
  oneof kind {
    LineEntity line = 1;
    // And other entity types.
  }
}
```

```
message Object {
  oneof kind {
    Entity entity = 1;
    Constraint constraint = 2;
  }
}
```

```
message Constraint {
  // Defined similarly to Entity.
}
```

```
message Sketch {
  repeated Object objects = 1;
}
```

(b) **A full sketch** is defined as a sequence of objects each of which can be either an entity or a constraint.

Listing 1: **Examples of object specifications.** We represent objects using *Protocol Buffers*. Protocol Buffers allow us to easily write specifications for structured objects of varying complexity.

Our ultimate goal is to build a machine learning model of sketch objects. To that end, we process the data even further and represent these objects as sequences of *tokens*. This allows us to pose sketch generation as *language modeling* (LM) and take advantage of the recent progress in this area [26, 6]. To achieve this, we pack first collections of entities and constraints into one Protocol Buffer message (see Listing 1b) assuming *some* ordering of objects. We discuss different orderings in Section 5.

There are a few ways to obtaining a sequence of tokens from a sketch message. Arguably the most intuitive one is to format messages as text. For a line entity connecting $(0.0, 0.1)$ and $(-0.5, 0.2)$ this will result in:

```
{ is_construction: true, start { x: 0.0, y: 0.1 }, end { x: -0.5, y: 0.2 } }
```

Since this format contains both the structure and the content of the data, the resulting sequences end up being prohibitively long. Additionally, the model would have to generate valid syntax, which would take up some portion of the model's capacity. To overcome these challenges, we work with two flavours of serialized PB messages.

The first one is a *sequences of bytes* obtained by calling the `SerializeToString()` method of a message. Such sequences are much shorter since the structure is handled by an external *parser* automatically generated from the data specification. The parser's task is to *interpret* the incoming stream of unstructured bytes and populate the fields of PB messages. However, like the text format, not every sequence of bytes results in a valid PB message.

Going one step further, we can utilize the structure of the sketch format more directly, and build a custom interpreter, that takes as input a sequence of tokens each representing a valid choice at various decision steps [4] in the sketch creation process. We designed this interpreter in such a way that all sequences of tokens in this format lead to valid PB messages. More specifically, we represent a message as a sequence of *triplets* $(d_i, c_i, f_i)$ where $i$ is an index of the token. The majority of tokens describe basic fields of the sketch objects with each token representing exactly one field. The first two positions in each triplet are allocated for a *discrete value* and a *continuous value* respectively. Since each field in a message is either discrete or continuous only one of two positions is *active* at a time (the other one is set to a default zero value). The third component is a *boolean flag* signifying the end of a `repeated` field[3] which contains a list of elements of the same type. An example sequence for a sketch containing a line and a point placed at one of its ends is shown in Table 1 (Triplet column).

---

[3] For additional details on how we handle more complex constructions of the PB language see Appendix A.

| | Triplet | Field | | | Triplet | Field | |
|---|---|---|---|---|---|---|---|
| 1. | $(\mathbf{0}, 0.0, \text{False})$ | `objects.kind` | | 8. | $(\mathbf{0}, 0.0, \text{False})$ | `objects.kind` | |
| 2. | $(\mathbf{0}, 0.0, \text{False})$ | `entity.kind` | | 9. | $(\mathbf{1}, 0.0, \text{False})$ | `entity.kind` | |
| 3. | $(\mathbf{1}, 0.0, \text{False})$ | `line.is_constr` | Line | 10. | $(\mathbf{0}, 0.0, \text{False})$ | `point.is_const` | Point |
| 4. | $(0, \mathbf{0.0}, \text{False})$ | `line.start.x` | | 11. | $(0, \mathbf{0.0}, \text{False})$ | `point.x` | |
| 5. | $(0, \mathbf{0.1}, \text{False})$ | `line.start.y` | | 12. | $(0, \mathbf{0.1}, \text{False})$ | `point.y` | |
| 6. | $(0, \mathbf{-0.5}, \text{False})$ | `line.end.x` | | 13. | $(0, 0.0, \mathbf{True})$ | `objects.kind` | |
| 7. | $(0, \mathbf{0.2}, \text{False})$ | `line.end.y` | | | | | |

Table 1: **A triplet representation of a simple sketch.** The sketch contains and a line and a point. Within each triplet in the *left column*, the *active* component (the value that is actually used) is highlighted in **bold**. The *right column* shows which field of the object the triplet is associated with.

Given a sequence of such triplets, it is possible to infer which exact field each token corresponds to. Indeed, the very first token $(d_1, c_1, f_1)$ is always associated with `objects.kind` since it is the first choice that needs to be made to create a `Sketch` message (see Listing 1b). The second field depends on the concrete value of $d_1$. If $d_1 = 0$ then the first object is an `entity` which means that the second token corresponds to `entity.kind`. The rest of the sequence is associated in a similar fashion. Field identifiers along with their locations within an object form the *context* of the tokens. We use this contextual information as an additional input for our machine learning models since it makes it easier to interpret the meaning of the triplet values and to be aware of the overall structure of the data.

## 4 Model

In order to estimate the distribution $p_{\text{data}}$ of 2D sketches in a dataset $\mathcal{D}$, we decompose the joint distribution over the sequence of tokens [19] $\mathbf{t} = (t_1, \dots, t_N)$ in an autoregressive fashion, representing each conditional with a neural network parameterized by $\theta$ and pose the estimation of $p_{\text{data}}$ as maximization of the log-likelihood of $\mathcal{D}$, *i.e.*,

$$p(\mathbf{t}; \theta) = \prod_{i=1}^{N} p(t_i \mid t_{<i}; \theta), \quad \sum_{\mathbf{t} \in \mathcal{D}} \log p(\mathbf{t}; \theta) \to \max_{\theta}, \tag{1}$$

where $N$ is the length of the sequence and $t_{<i}$ denotes all the tokens preceding $t_i$.

More concretely, we employ the *Transformer decoder* architecture [33] that takes an embedding of the token $\boldsymbol{e}_{i-1} = \texttt{embed}_i(t_{i-1}) \in \mathbb{R}^D$ and maps it into another vector $\boldsymbol{h}_i$ of the same dimensionality. The latter is decoded into parameters of $p(t_i \mid t_{<i})$ by a learned mapping $\texttt{dist}_i(\cdot)$.[4]

**Byte representation** When dealing with the bytes of a PB message, each token is simply a discrete value in the range $\{0, \dots, 255\} \cup \{\texttt{EOS}\}$ and therefore $p(t_i \mid t_{<i}; \theta)$ can be modeled as a *categorical* distribution similar to how it's done in typical LM approaches [5]. In this setting, for each time step $i$ of the sequence we have

$$\texttt{embed}_i(t_{i-1}) = V[t_{i-1}] + \boldsymbol{e}_i^{\text{pos}}, \tag{2}$$

where $[\cdot]$ denotes the lookup operation and $\boldsymbol{e}_i^{\text{pos}}$ is a position embedding for position $i$. Both $V$ and $\boldsymbol{e}_i^{\text{pos}}$ are learned. Moreover, $\forall i\ \texttt{dist}_i$ is the same linear projection into $\mathbb{R}^{257}$ (256 values and `EOS`) and the output is treated as *logits* of the distribution.

**Triplet representation** In case of the triplet representation, we follow a slightly more involved procedure. As outlined in Section 3, tokens can be either discrete or continuous. Additionally, different discrete tokens may have *different ranges of values*. For example, there are only two possible values for the `object.kind` token – either to an entity or a constraint. On the other hand, the range of the `entity.kind` token has cardinality of 4 since we support 4 different types of sketch entities. This means that we can't naively describe each conditional in Equation 1 using the same template distribution like we did for the bytes. We circumvent this by introducing the notion of *token groups*.

A token group $\mathcal{G}$ is a collection of related token types that can be handled in a similar fashion. Specifically, we use the same embedding function and the same projection for every $t$ such that

---

[4]Note that we use subscript $i$ for `embed` and `dist` since they can be different for different time steps.

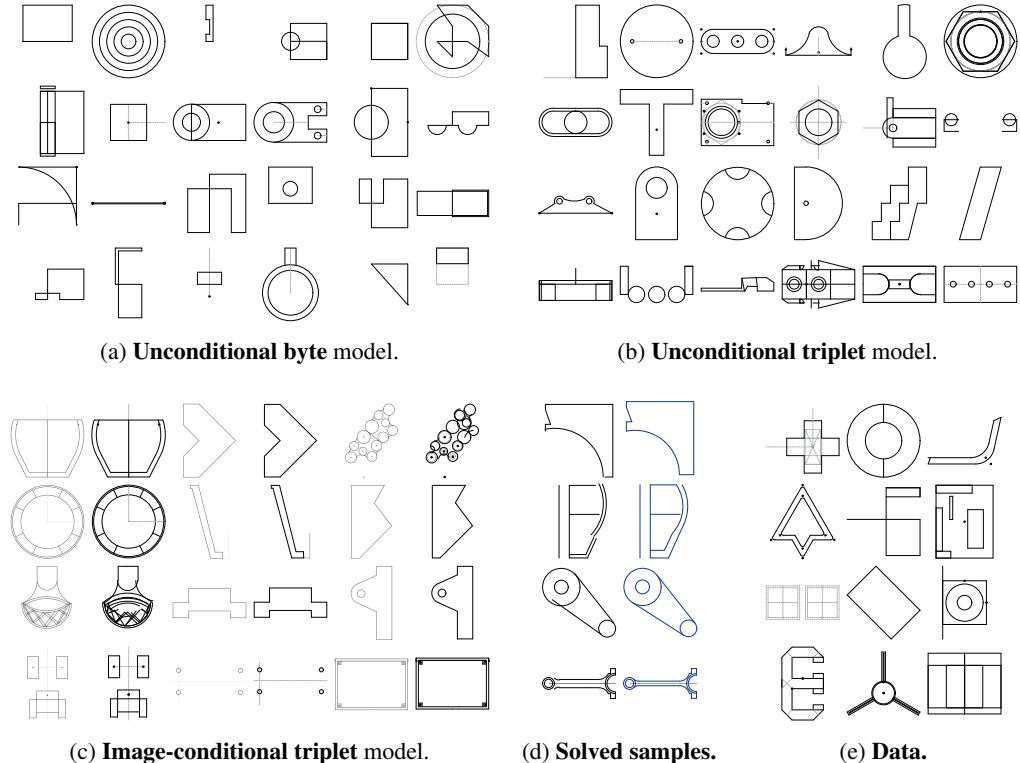

(a) **Unconditional byte** model.

(b) **Unconditional triplet** model.

(c) **Image-conditional triplet** model.

(d) **Solved samples.**

(e) **Data.**

Figure 3: **Synthesized sketches and data. (a)–(c)** show samples from various proposed models (we use Nucleus Sampling with *top-p* = 0.9). In **(c)**, the output is rendered in a slightly thicker style. **(d)** demonstrates samples from the unconditional model after applying predicted constraints (the output is in *blue*). **(e)** shows examples from the dataset.

$\mathtt{type}(t) \in \mathcal{G}$. For instance, we might want to group all the tokens associated with coordinates. In the example from Table 1, tokens with indices 4–7 and 11–12 will all end up in the same $\mathcal{G}$[5]. Naturally, we use the same functional form for the output distribution $p(t_i \mid t_{<i}; \theta)$ within each group.

We embed $t_{i-1}$ that belongs to the group $\mathcal{G}$ as (note the difference with Equation 2):

$$K[\mathtt{field}(t_{i-1})] + V^{\mathcal{G}}[t_{i-1}] + \boldsymbol{e}_n^{\mathrm{obj}} + \boldsymbol{e}_m^{\mathrm{rel}} ,\qquad(3)$$

where $\mathtt{field}(t)$ returns the field of $t$ (*e.g.*, `objects.kind` or `line.start.x`) and $K$ is a collection of learnable embeddings for every possible field type. Unlike in Equation 2, instead of using global position embedding $\boldsymbol{e}_i^{\mathrm{pos}}$ we describe the location with the index $n$ of the current object as well as the relative position $m$ of $t_{i-1}$ within an object.

We handle the "end" tokens (*i.e.*, $f_i$ = True) similarly to EOS in Section 4 — the output projection produces an additional logit used to compute the probability of ending the repetition. Since $f_i$ is only expected to be True at particular points in the sequence (*i.e.*, right after tokens forming a *whole* element of the list) we *mask out* the extra logit everywhere else. This ensures that the "end" token can't be predicted prematurely and also eliminates its unnecessary contribution to the optimized objective.

One significant difference between the byte setting and the triplet setting is how we process pointer fields. In the former, pointers do not get any special treatment and are generated just like any other integer field. We rely on the model's capability to make sense of the entity part index and relate it to the corresponding locations in the sequence via attention weights. This seems to be a viable strategy since Transformers have demonstrated an impressive referencing capacity in recent works [6].

Since the triplet representation provides us with direct access to the semantics of tokens it's possible to relate pointers to their pointees more explicitly by using Pointer Networks [35]. The approach we

---

[5]For the full list of token groups used in the model see Appendix B.

are taking here is similar to [21]. In order to compute $p(t_i \mid t_{<i}; \theta)$, we first project the output of the Transformer $\boldsymbol{h}_i$ into the final pointer vector $\boldsymbol{p}_i = W_{\text{ptr}}\boldsymbol{h}_i$. The conditional is then obtained as follows:

$$p(t_i \text{ points to } t_{ik} \mid t_{<i}; \theta) = \text{softmax}_k(\boldsymbol{p}_i^T H_i), \quad H_i = [\boldsymbol{h}_{i1}, \dots, \boldsymbol{h}_{ik}, \dots, \boldsymbol{h}_{iM}], \quad (4)$$

where $\text{softmax}_k$ is the $k$-th element of the softmax vector and $\boldsymbol{r}_i = \{t_{i1}, \dots, t_{iM}\}$ is the set of tokens that $t_i$ can point to. Naturally, $\boldsymbol{r}_i$ only contains tokens from $t_{<i}$. This is different from [35, 21] where $\boldsymbol{r}_i \equiv \boldsymbol{r}$ is external to the predicted sequence and remains immutable throughout the generation process.

In Onshape, constraints can be applied not only to whole entities but also to their subparts (*e.g.*, the center of a circle or the end points of a line segment). To handle this, we introduce special *referrable tokens* decoupling pointees from the object attribute tokens. Referrables are injected after each entity and have the same identifier within each entity type. In order to let the model distinguish between different subparts, we adjust Equation 3 to use a learnable embedding of a part index instead of $V^{\mathcal{G}}[t_{i-1}]$. As referrable tokens do not need to be predicted the respective terms are removed from Equation 1. For the same reason, in Equation 4, $\boldsymbol{h}_{ik}$ corresponds to the time step where $t_{ik}$ is the *output* and not the input. This allows us to avoid having unused Transformer outputs.

Finally, we need to specify how we embed pointers as inputs to the Transformer network. Following [35, 21] we could reuse $\boldsymbol{h}_j$ for tokens that point to $t_j$. Unfortunately, this creates output-to-input connections which are extremely detrimental to the efficiency of the Transformer architecture — different time steps can no longer be processed in parallel during training. Instead, we opt for a simpler solution and employ the standard embedding scheme for discrete tokens (Equation 3).

**Sampling from the model**   Sampling from the byte model is identical to sampling from any typical Transformer-based LM. The triplet model, on the other hand, requires slightly more bespoke handling. Figure 2 illustrates the procedure. We start by embedding and feeding a special BOS token into the Transformer. The Transformer then outputs a collection of triplets, one for each possible token group. In order to determine which concrete token needs to be emitted, we employ an interpreter (a state machine) automatically generated from the data specification. Knowing the current state allows us to choose the right token group and associate the active component of the triplet with a field in the synthesized object. Once the appropriate field is populated the interpreter transitions to the next state and produces an output token which is then fed back into the model. The process stops when the state machine receives the "end" triplet for the outermost repeated field (*i.e.*, `object.kind`).

**Conditional generation**   In addition to the unconditional model described above, we explore a variant that allows us to translate bitmaps into sketches. Here, we simply let the main Transformer cross-attend to the features extracted from the input image by a ViT network [12]. The specific setup is detailed in Appendix C.

## 5   Experiments

We validate our proposed approaches on the data that we obtained from the repository of documents publicly available on the Onshape platform [22]. Following the standard evaluation methodology for autoregressive generative models [23, 21] we use log-likelihood as our primary quantitative metric. Additionally, we provide a variety of random and selected model samples for qualitative assessment (Figure 3).

**Dataset**   Unlike the majority of the existing works dealing with CAD sketch generation [31, 37, 24] we do not rely on SketchGraphs [31]. Instead, we collect the largest to date dataset of engineering sketches addressing the main disadvantage of [31], severe data duplication. The acquisition and the filtering procedures

| Model | Sequence | Average bits per | |
| --- | --- | --- | --- |
| | | object | sketch |
| Uniform | unord. bytes | 112.23 | 3683.56 |
| | unord. triplets | 25.34 | 847.52 |
| Text | interleaved | 4.622 | 139.687 |
| Byte | concatenated | 4.381 | 132.621 |
| | interleaved | 4.252 | 127.495 |
| Triplet | concatenated | 4.218 | 127.913 |
| | interleaved | 4.103 | 123.213 |
| Cond. | interleaved | 1.570 | 53.730 |

Table 2: **Test likelihoods of various models.** The *object* column is computed as the average number of bits per object in a sketch averaged across test examples. The *sketch* column is similar except we do not divide by the number of objects.

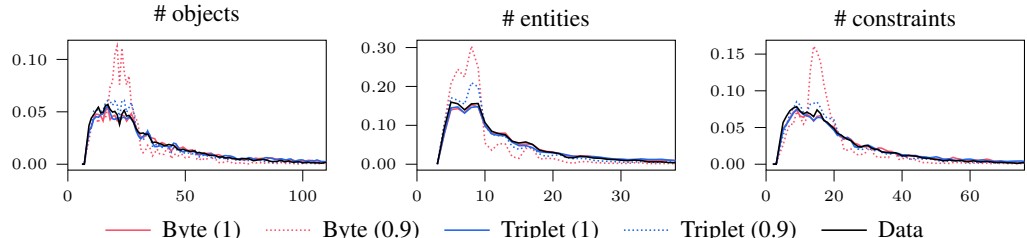

Figure 4: **Distribution of various sketch statistics** for samples drawn from our unconditional models. The *top-p* parameter for Nucleus Sampling is shown in parentheses.

are detailed in Appendix D. For our experiments we split the dataset randomly into 3 parts: 4,656,607 examples for the training set and 50,000 sketches for each the validation and the test set.[6]

**Unconditional generation.** In this series of experiments, the goal is to determine how well our models capture the distribution of sketches in the dataset. We use the same network architecture for both the byte and the triplet settings.[7] We compare two orderings of objects: in the first one (*concatenated*), constraints go after the last entity while in the second (*interleaved*), a constraint object is injected immediately after the entities it operates on. In both cases, the relative orderings within both the sequence of entities and the sequence of constraints are taken directly from the original JSON messages.

Table 2 shows test log-likelihoods obtained by different models. Unsurprisingly, our proposed methods (rows 4–7) significantly outperform the weak baselines (rows 1–2). The difference between the two uniform settings is due to the fact that the byte description of a PB message is usually longer than the triplet one: 239 *vs.* 456 tokens on average with the maximum length of 959 *vs.* 1987. Additionally, the tokens in the triplet representation tend to have smaller range of values (*i.e.*, $< 257$).

These differences between representations may partially explain why triplet models demonstrate better performance on the hold-out test set. It's also worth emphasizing that the byte model does not receive any explicit information about the parsing state. This seem to make learning more challenging and as a result compared to the triplet model it takes roughly 3 times more network updates to reach the highest data likelihood on the validation.

Another important factor affecting the performance of the models is the choice of the object ordering. As it is evident from Table 2, the interleaved ordering consistently leads to better results. One explanation for this is that at any point in time, the model has more explicit information about the relations between the sketch entities produced so far.

In the row 3 of Table 2, we also provide the test log-likelihood for a conventional language modeling baseline trained on the text representation of PB messages (see Section 3). Here, we employ the SentencePiece tokenizer [17] with fairly aggressive settings (8000 words in the vocabulary; splitting at whitespaces switched off) to be able to keep the sequences within the budget of 1024 tokens. As can be seen from the table, the resulting Transformer model (of exactly the same architecture as other entries) despite being better than the uniform baselines is still substantially worse than the proposed methods.

In addition to measuring likelihoods, we sampled 10,000 sketches from the best performing byte and triplet models and computed distributions of various high-level statistics (Figure 4 and Figure 6 in the appendix). We repeated this procedure both with and without using Nucleus Sampling (NS) [15]. Both models follow the data distribution closely when we use samples from the unmodified model output. In this setting, however, a significant fraction of sketches is either malformed (*e.g.*, the generated PB message cannot be parsed) or unsolvable: 36% for the byte model and 14% for the triplet model. NS with *top-p* $= 0.9$ skews the sample distribution and seems to have a more pronounced negative effect on the byte model. The upside is that the resulting sketches become "cleaner": the percentage of invalid samples goes down to 25% and 6% for the byte and the triplet settings respectively.

---

[6]The dataset is available at `https://bit.ly/3m9QHPd`.

[7]Please refer to Appendix E for details.

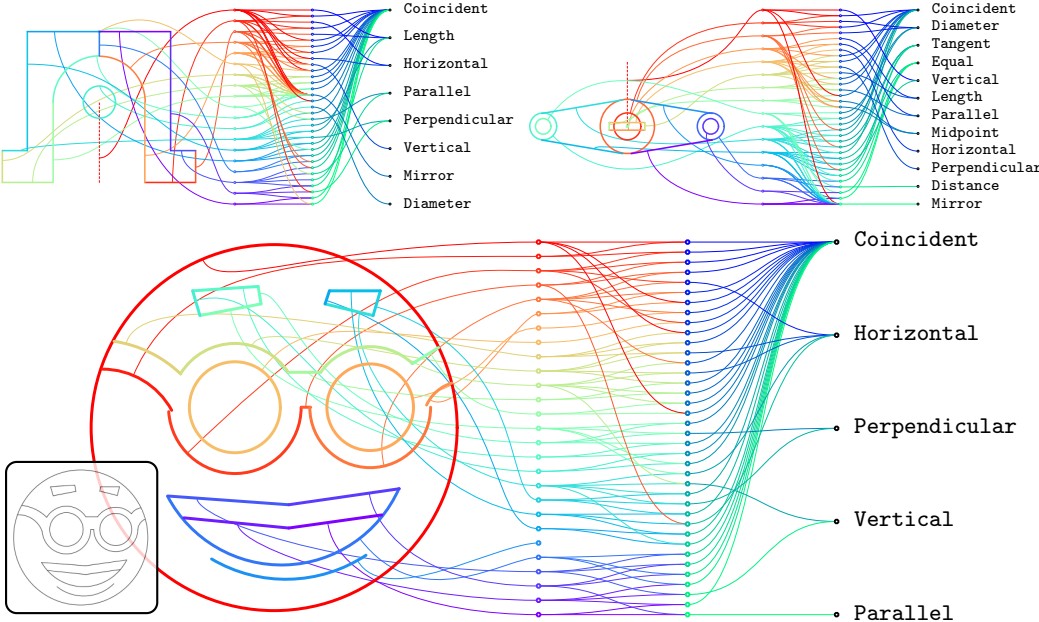

Figure 5: **Entities and constraints** sampled from the unconditional (*top*) and the conditional (*bottom*) triplet models. The *first column* of nodes represents different entities (all parts are folded into a single node). The order of nodes (top to bottom) follows the generation order. The *second column* represents different constraints also ordered by their index in the sequence. Finally, the *third column* is reserved for constraint types, from the most to the least frequent.

Figures 3a and 3b show renders of random samples from several proposed models. Overall, generated sketches look plausible and exhibit a lot of desired properties: closedness of regions, regularity, symmetry, a non-trivial amount of fine detail. We observe that the byte model produces slightly less complex samples with fewer open arcs but this could be a side effect of a particular *top-p* value. We also note that the model does not always synthesize sensible sketches — just like any other typical autoregressive model trained with *teacher forcing* it suffers from not being able to recover from mistakes made early on in the sequence [28]. This can potentially be addressed by fine-tuning using, for example, reinforcement learning.

We show a more detailed sample from the triplet model in Figure 5. While not perfect, the inferred constraints are reasonable most of the time. The model does a good job at connecting entities using `Coincident` constraints but also successfully detects more complex relations spanning more than two primitives (*e.g.*, `Mirror`). Having access to constraints gives us opportunity to correct mistakes in entity prediction by applying an external sketch solver (see Figure 3d). Although this aspect of sketching was not the main focus of this work, we believe that a tighter integration between the model and the CAD software will lead to a significant boost in generation quality.

**Conditional generation.** As discussed in Section 4, we also trained an image-conditional model using the same regime as for the unconditional one. As expected, it achieves a significantly better fit (the last row of Table 2) but at the same time retains a non-trivial amount of uncertainty. The latter is arising, in particular, from the fact that different permutations of entities result in the same rendered image. Image-conditional samples can be found in Figure 3c. The model was able to nearly perfectly reconstruct simpler sketches and mostly made mistakes in the presence of a large number of fine details. Additionally, we compared our system against the nearest neighbour baseline in terms of visual reconstruction error as measured by Chamfer distance. We found that the proposed method reduces the error by $\approx 80\%$ (see Appendix H). In order to test the out-of-distribution performance, we supply the model with several custom-made drawings. Surprisingly, after minor post-processing to account for the short sequence bias the system manages to produce reasonable reconstructions (see the bottom example in Figure 5). We detail this experiment in Appendix I.

# 6 Discussion

In this work, we have demonstrated how a combination of a general-purpose language modeling technique alongside an off-the-shelf data serialization protocol can be used to effectively solve generation of complex structured objects. We showcased the proposed system on the domain of 2D CAD drawings and developed models that can synthesize geometric primitives and relations between them both unconditionally as well as using a bitmap as a reference. These are only initial proof-of-concept experiments and we are hoping to see more applications taking advantage of the flexibility of the developed interface: conditioning on various sketch properties, inferring constraints given entities and automatically completing drawings, to name a few.

Although we focused our attention on a particular dataset we argue that the approach described in this paper is largely domain-agnostic. In order to adapt the system to a new kind of data, the algorithm designer only needs to provide an appropriate Protocol Buffer specification and if the PB language is too restrictive one can always replace it with a more powerful interpreter. As a straightforward direction for future work, we can consider extending the method to handle 3D. In Onshape, 3D operations bear a lot of similarities with sketch constraints — just like constraints, they can be represented as nested messages containing references to the geometries existing in the scene. Thus, most of the ideas from the present work can be taken verbatim to this new setting.

We hope that this work will serve as a stepping stone for further advances in the field of automated CAD but also inspire new ideas and approaches to generative modeling of arbitrary structured data.

## Acknowledgments and Disclosure of Funding

The authors would like to thank Charlie Nash, Georg Ostrovski, and Adam Kosiorek for helping with the manuscript preparation as well as Igor Babuschkin, David Choi, Nate Kushman, Andrew Kimpton, Jake Rosenfeld, Lana Saksonov, John Rousseau, Greg Guarriello, Andy Brock, Francesco Nori, Aäron van den Oord, and Oriol Vinyals for insightful discussions and support.

YG, SB, YL, and SS are funded by DeepMind. EK is funded by PTC.

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
