# A    Object specifications

This section contains additional details on the object specifications. As mentioned in Section 3, we rely on the PB language to define the structure for each object type that we would like to handle with our model. Our framework supports all basic constructions of the language including nested messages and `oneof` clauses.

The `oneof` clause marks mutually exclusive fields. For example, in Listing 1b, we can see that a generic `Object` can be either an `entity` or a `constraint`. We also use `oneof` for objects that may appear in several mutually exclusive configurations (*e.g.*, `CircleArcEntity` represents both arcs and closed circles and for the latter which it does not make sense to specify end points). We handle such constructions by injecting an additional token with the discrete value set to the index of the active field. In some cases, the configuration is determined by a group of preceding field values. In situations like this, the `oneof` branch is chosen by a special *handler* that observes the fields populated so far. This handler is passed as an *option* to the construction (see `DistanceConstraint` definition in Listing 3).

Another feature of the PB language that is essential for defining sketches is `repeated` fields. A repeated field is a list of an arbitrary number of elements of the same type. We first see this used in the only top-level field of the `Sketch` message (Listing 1b): a sketch is a list of `Objects`. In order to represent such fields, we could just concatenate the tokens of all the elements. The only problem we will face is during generation when the model needs to indicate that it has finished producing the list. Specifically for this situation, we use the third position of the token triplet, $f_i$. It is a *boolean flag* that signifies the end of repetition. Each "end" token is set to $(0, 0.0, \textbf{True})$. Note that since messages may have nested repeated fields the resulting sequence may contain several "end" tokens. For some repeated fields we also specify the minimum number of items that needs to be generated (*e.g.*, constraints should refer to at least one entity). In the PB specification, such fields are marked with the `at_least` field option.

In Listings 2 and 3, we are showing all the supported object specifications. Although most of them are direct adaptations of the JSON structures returned by the Onshape API we made an effort to remove redundant and irrelevant sections. We also attempted to bring our constraint specifications closer to how they behave in the software UI. For instance, the *coincident constraint* can usually be applied to several entities at once. In the API, however, this is translated into a collection of pairwise constraints. In our dataset, we compress this collection back into a single object.

The handler in `DistanceConstraint` is defined as a `python` function:

```python
def select_distance_params(obj):
  if obj.direction in {"HORIZONTAL",
                       "VERTICAL"}:
    return 0
  else:
    return 1
```

# B    Token groups

Table 3 details the specific grouping of tokens used in our experiments. Groups are defined by regular expressions matching subsets of dotted field identifiers. For each group we provide the range of possible values.

# C    Conditional generation

In addition to the model described above that generates sketches from scratch, we explore a modification that allows generation of a sketch based on its visual render or drawing. The conditional model relies on the same transformer-based architecture with an additional input sequence $H^{\text{img}} = [\mathbf{h}_1^{\text{img}}, \ldots, \mathbf{h}_P^{\text{img}}]$ to which the transformer can attend at every layer. We obtain $H^{\text{img}}$ by embedding the conditioning image using visual transformer [12] which extracts patches of a certain size from the image and processes them by a number of self-attention layers.

```
message Vector {
  double x = 1;
  double y = 2;                          message LineEntity {
}                                          bool is_construction = 1;
                                           Vector start = 2;
message PointEntity {                      Vector end = 3;
  bool is_construction = 1;            }
  Vector point = 2;
}                                        message InterpolatedSplineEntity {
                                           bool is_construction = 1;
message CircleArcEntity {                  bool is_periodic = 2;
  bool is_construction = 1;                repeated Vector interp_points = 3
  Vector center = 2;                           [(field_options).at_least = 2];
  message CircleParams {                   Vector start_derivative = 4;
    double radius = 1;                     Vector end_derivative = 5;
  }                                        message TrimmedParams {
  message ArcParams {                        double start_phi = 1;
    Vector start = 1;                        double end_phi = 2;
    Vector end = 2;                        }
    bool is_clockwise = 3;                 oneof additional_params {
  }                                          Empty untrimmed_params = 6;
  oneof additional_params {                  TrimmedParams trimmed_params = 7;
    CircleParams circle_params = 3;        }
    ArcParams arc_params = 4;            }
  }
}
```

Listing 2: Protocol Buffer specifications for the supported **sketch entities**.

This scheme not only allowed us to conveniently contain the conditional model within the same transformer framework, but also allowed the generator to attend only to relevant parts of the image when processing each of the objects. This proved to be crucial for reconstructing finer details of the sketch and could not be achieved with a more traditional single-vector representation of the image.

We train the conditional model in a manner similar to Equation 1:

$$\theta^* = \arg\max_\theta \sum_{t \in \mathcal{D}} \log p(\mathbf{t}|\text{image}(\mathbf{t}); \theta), \tag{5}$$

where $\text{image}(\mathbf{t})$ is a computer render for the token sequence $\mathbf{t}$ and $\theta$ now includes parameters of the visual transformer too. Whenever a parallel dataset is available consisting of sketches and their drawings (e.g. human-drawn), the model can be trained accordingly which we leave for future work.

## D   Dataset

The data we rely on in the work originates from the Onshape website [22]. The website contains a big collection of CAD models which can be used freely for non-commercial purposes. Because of this there have been several attempts to employ Onshape as a source of research data [16, 31]. Unfortunately, in the context of CAD sketches, all the available datasets derived from the Onshape public repository seem to have limitations. In particular, the Onshape API makes it difficult to obtain raw parameters of sketch entities and constraints since they are often defined as symbolic expressions with variables. These variables may in turn correspond to expressions that are specified elsewhere in the document and so on. The simplest way to deal with this is to just ignore such sketches and only consider instances relying on plain values. Prior efforts take this route and therefore discard a significant portion of the available data.

In order to collect a more complete dataset of sketches, we performed a large-scale scrape downloading around 6TB of public documents in October, 2020. We only consider sketches containing the most common entity and constraint types (4 and 16 different types respectively). Additionally, we remove constraints that refer to so-called "external" entities, *i.e.*, elements of the CAD model that exist outside the sketch plane.

```protobuf
message CoincidentConstraint {
  repeated Pointer entities = 1
      [(field_options).at_least = 2];
}

message ConcentricConstraint {
  repeated Pointer entities = 1
      [(field_options).at_least = 2];
}

message EqualConstraint {
  repeated Pointer entities = 1
      [(field_options).at_least = 2];
}

message ParallelConstraint {
  repeated Pointer entities = 1
      [(field_options).at_least = 2];
}

message TangentConstraint {
  Pointer first = 1;
  Pointer second = 2;
}

message PerpendicularConstraint {
  Pointer first = 1;
  Pointer second = 2;
}

message MirrorConstraint {
  Pointer mirror = 1;
  message Pair {
    Pointer first = 1;
    Pointer second = 2;
  }
  repeated Pair mirrored_pairs = 2
      [(field_options).at_least = 1];
}

message LengthConstraint {
  Pointer entity = 1;
  double length = 2;
}

message DiameterConstraint {
  Pointer entity = 1;
  double length = 2;
}

message RadiusConstraint {
  Pointer entity = 1;
  double length = 2;
}

message AngleConstraint {
  Pointer first = 1;
  Pointer second = 2;
  double angle = 3;
}

message FixConstraint {
  repeated Pointer entities = 1
      [(field_options).at_least = 1];
}

message DistanceConstraint {
  Pointer first = 1;
  Pointer second = 2;
  enum Direction {
    HORIZONTAL = 0;
    VERTICAL = 1;
    MINIMUM = 2;
  }
  Direction direction = 3;
  double length = 4;
  enum Alignment {
    ALIGNED = 0;
    ANTI_ALIGNED = 1;
  }
  enum HalfSpace {
    NOT_AVAILABLE = 0;
    LEFT = 1;
    RIGHT = 2;
  }
  message HalfSpaceParams {
    HalfSpace half_space_first = 1;
    HalfSpace half_space_second = 2;
  }
  oneof additional_params {
    option (oneof_options).handler =
        "select_distance_params";
    // Defined further in the text.

    Alignment alignment = 5;
    HalfSpaceParams
        half_space_params = 6;
  }
}

message HorizontalConstraint {
  repeated Pointer entities = 1
      [(field_options).at_least = 1];
}

message VerticalConstraint {
  repeated Pointer entities = 2
      [(field_options).at_least = 1];
}

message MidpointConstraint {
  Pointer midpoint = 1;
  message Endpoints {
    Pointer first = 1;
    Pointer second = 2;
  }
  oneof additional_params {
    Endpoints endpoints = 2;
    Pointer entity = 3;
  }
}
```

Listing 3: Protocol Buffer specifications for the supported **sketch constraints**.

| | Regular expression | Range |
|---|---|---|
| Discrete tokens | | |
| | `objects\.kind` | $\{0, 1\}$ |
| | `.*\.entity.kind` | $\{0, 1, 2, 3\}$ |
| | `.*\.constraint.kind` | $\{0, \ldots, 15\}$ |
| | `.*\.is_construction` | $\{0, 1\}$ |
| | `.*\.is_clockwise` | $\{0, 1\}$ |
| | `.*\.is_periodic` | $\{0, 1\}$ |
| | `.*\.additional_params` | $\{0, 1\}$ |
| | `.*\.(entity|entities|first|second|midpoint|mirror)` | $\{0, \ldots, 255\}$ |
| | `.*\.direction` | $\{0, 1, 2\}$ |
| | `.*\.alignment` | $\{0, 1\}$ |
| | `.*\.half_space_(first|second)` | $\{0, 1, 2\}$ |
| Continuous tokens (quantized to 8 bits) | | |
| | `.*\.(point|start|end|center|interpolation_points)\.(x|y)` | $[-1.0, 1.0]$ |
| | `.*\.(start|end)_derivative\.(x|y)` | $[-100.0, 100.0]$ |
| | `.*\.(start|end)_phi` | $[0.0, 3.0]$ |
| | `.*\.radius` | $[0.0, 1.0]$ |
| | `.*\.length` | $[0.0, 2\sqrt{2}]$ |
| | `.*\.angle` | $[-10.0, 10.0]$ |

Table 3: **Token groups** used in our experiments. The *left column* contains `python`-style regular expressions isolating specific subsets of tokens (*i.e.*, dotted paths of the corresponding fields). The *right column* shows the ranges of values for each group.

We then preprocess the remaining data to replace symbolic expressions with plain values. To that end, we first employ the *default configuration* file to obtain the initial assignments of variables and then scan through the document locating further assignments and updating the variable mapping. If an assignment contains a symbolic expression we evaluate it using SymPy[8] [18]. Once the scan is finished we can use plain values of the variables to resolve any expressions in the sketch parameters.

CAD models in the public repository cover a wide range of objects in the real world, anything from coffee mugs to sports cars. As a result, sketches have significant variability in size. To make training of our models easier, we rescale examples in the dataset to lie in the $[-1, 1] \times [-1, 1]$ bounding box.

Another topic that is somewhat overlooked in [31, 16] is duplicate data. As is the dataset ends up having many sketches that are either single built-in primitives or copied (sometimes with minor modifications) from popular documents or tutorials. Not only this restricts the diversity of samples generated by machine learning models but also makes assessment of such models difficult – a random subset chosen for testing is likely to significantly overlap with the training set.

We make a best-effort attempt to address duplication by using the following filtering procedure. We first remove simple axis-aligned rectangles which constitute around $\approx 15\%$ of the data. Next, we divide the remaining examples into bins of *semantically equivalent* sketches. Two sketches are considered to be equivalent if they share the same sequence of *object types* (*e.g.*, a line followed by a point followed by a coincidence constraint). For more complex examples that have matching sequences of types there is a good chance that the examples themselves look very similar.

---

[8]This works most of the time but there are rare cases when SymPy fails to parse Onshape's syntax.

The resulting bins contain manageable numbers of sketches (up to several tens of thousands) and therefore can undergo more computationally expensive filtering. Within each bin, we obtain $128 \times 128$ binary renders of each example and apply hierarchical clustering[9] with 0.1 threshold and Jaccard distance as metric. The final dataset is formed by taking a single sketch per cluster (unsurprisingly, sketches from the same cluster end up looking almost identical).

For our experiments, we exclude the sketches that have fewer than 4 or more than 100 entities.

# E    Training details

We train our models for $10^6$ weight updates using batches with 128 rows. Each row can fit sequences up to 1024 tokens long in the triplet setting and 1990 tokens long in the byte setting. In order to improve occupancy and reduce wasted computation we fill up the rows dynamically packing as many examples as possible into a row before moving on to the next one. Each batch is processed in parallel by 32 TPU v3 cores.

We use the Adam optimizer with the learning rate of $10^{-4}$ and clip the gradient norm to 1.0. Additionally, we employ a dropout rate of 0.1 in all the experiments.

## E.1    Architecture details

For the unconditional models, we used embeddings and fully-connected layers of size 384 and 24 Transformer blocks. These sizes were manually selected by log-likelihood on the validation set from a small number of training runs. For the baseline, we employ a model that outputs uniform distribution at every time step.

In the image-conditional model we used 16 and 22 layers for the vision and triplet transformers correspondingly, both operating with 384-dimensional embeddings. We did not optimize these parameters, they were largely determined by the hardware memory limitations. For conditioning images we employed $256 \times 256$ sized renders divided into $16 \times 16$ image patches resulting in total $P = 256$ conditioning vectors $\mathbf{h}_p^{\text{img}}$. A brief experimentation with $8 \times 8$ and $32 \times 32$ sized patches suggested that $16 \times 16$ patch resolution is nearly optimal in terms of the memory consumption and model performance.

# F    Additional statistics

We present the distributions of a variety of additional sketch statistics in Figure 6. These results are discussed in Section 5.

# G    Compressing byte sequences

We tried making byte sequences shorter by feeding PB messages into a general purpose compression algorithm (Brotli [1] with the quality setting of 7). This allowed us to reduce the maximum length to 930 tokens, which is very similar to that of the triplet setting. Unfortunately, this change renders learning impossible – a model of the same capacity as before fails to improve beyond the initial reduction of the training objective. We suspect that this happens because the compression algorithm aggressively mangles the contents of a PB message and thus it is too difficult for the model to make sense of the data.

# H    Evaluation of the image-conditional model

In this appendix, we provide additional details on evaluation of the image-conditional version of our model. We measure how similar the inferred sketch is to the ground truth one in terms of resulting renders and number of entities and constraints. We also provide a simple yet meaningful baseline for the comparison which consists of finding a nearest neighbour in the train set for the queried test sketch in the space of their renders. We use the same distance metric both for evaluating the reconstruction

---

[9]We use `cluster.hierarchy.fclusterdata` from SciPy.

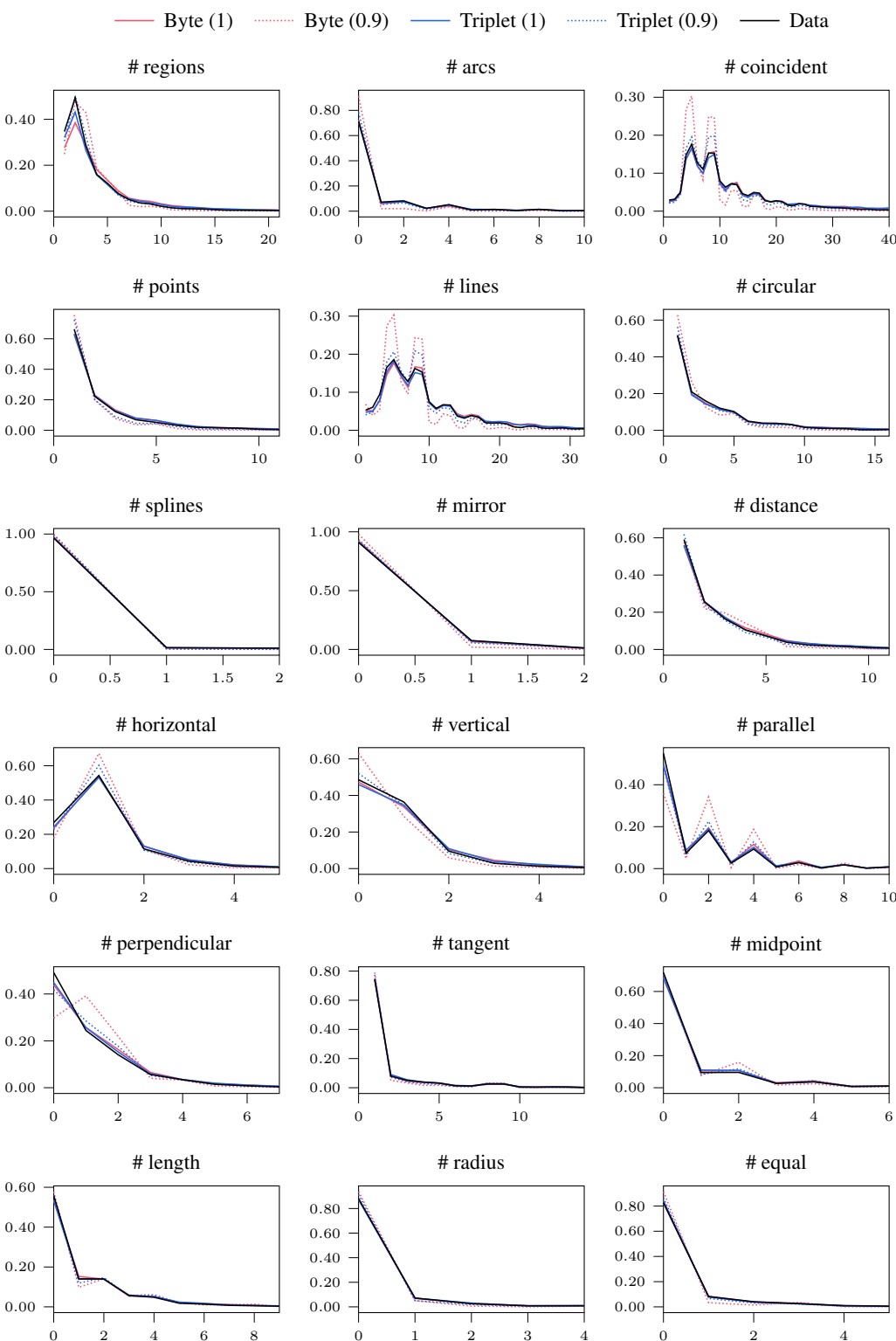

Figure 6: **Distribution of additional sketch statistics.** See also Figure 4.

| Metric | Baseline | Our model |
|---|---|---|
| Reconstruction error | 2.97 | **0.59** |
| # of entities discrepancy | 4.67 | **1.10** |
| # of constraints discrepancy | 9.82 | **5.33** |

Table 4: **Conditional model evaluation.** *Reconstruction error* is computed as Chamfer distance normalized over $256 \times 256$ pixels. *Discrepancy* in number of predicted entities and constraints is reported as absolute difference to the ground truth data. All metrics are averaged over the test set, see Appendix H for details.

error and finding nearest neighbours which is Chamfer distance [3] between binarized $256 \times 256$ renders. We found this metric to be a reasonable trade-off between simplicity and effectiveness in terms of detecting near or full duplicates. The quantitative comparison can be found in Table 4.

As expected, our model performs significantly better than the nearest neighbour baseline. Perhaps surprisingly, it also predicts a very close number of entities which may suggest the model captured the way a human would decompose the render into entities.

## I   Image-to-sketch translation for custom drawings

We discovered that the model when presented with renders of both test examples (*e.g.*, the top image in Figure 7) as well as custom-made drawings of medium complexity (*e.g.*, the second and the third row in Figure 7) is able predict sensible structured sketches. In more challenging situations, however, we observed that the generation process often stops prematurely producing only a subset of entities and constraints. In particular, the model struggled with a smiley face shown in Figure 5. We hypothesize that this is due to the fact that the majority of the training sequences contain only a small number of objects and therefore the model ends up being biased towards synthesizing shorter samples.

To overcome this arguably insignificant limitation we enforced the interpreter to continue generating tokens until the limit of 1024 tokens is reached. We then selected the longest subsequence from 64 sampled sequences that minimized the $L_2$ difference between the conditioning image and rendered result (after Gaussian smoothing with $\sigma = 4$), which can be seen as a form of model-guided search procedure. This procedure allowed us to obtain nearly perfect results for more complex input drawings like the aforementioned smiley face in Figure 5. One can note than even though the model did not place the arcs from which the eye-glasses are built very precisely, it still supplemented them with constraints that make sure their endpoints coincide. Besides this, mouth-forming lines drawn as almost parallel in the original sketch simply by the chance, were recognized as parallel in the provided sample with the corresponding constraints which is an example of good understanding of relationships between objects by the model.

Additionally, we conducted a preliminary experiment in which we trained a model on various image distortions (*e.g.*, adding slight noise to the entity parameters, warping and binarizing sketch renders) while keeping the target sketch untouched. This regime is meant to make the system more resilient to the quality of the input bitmap. The bottom example in Figure 7 demonstrates the performance of the resulting model on a scanned shape drawn by hand. It can be noted that although the output sketch is not perfect it does a good job at capturing the appearance of the reference image while presenting a plausible design intent.

## J   Additional samples

We show a collection of additional random samples from the unconditional triplet model in Figure 8. Like in Figure 3, we use Nucleus Sampling with *top-p* = 0.9.

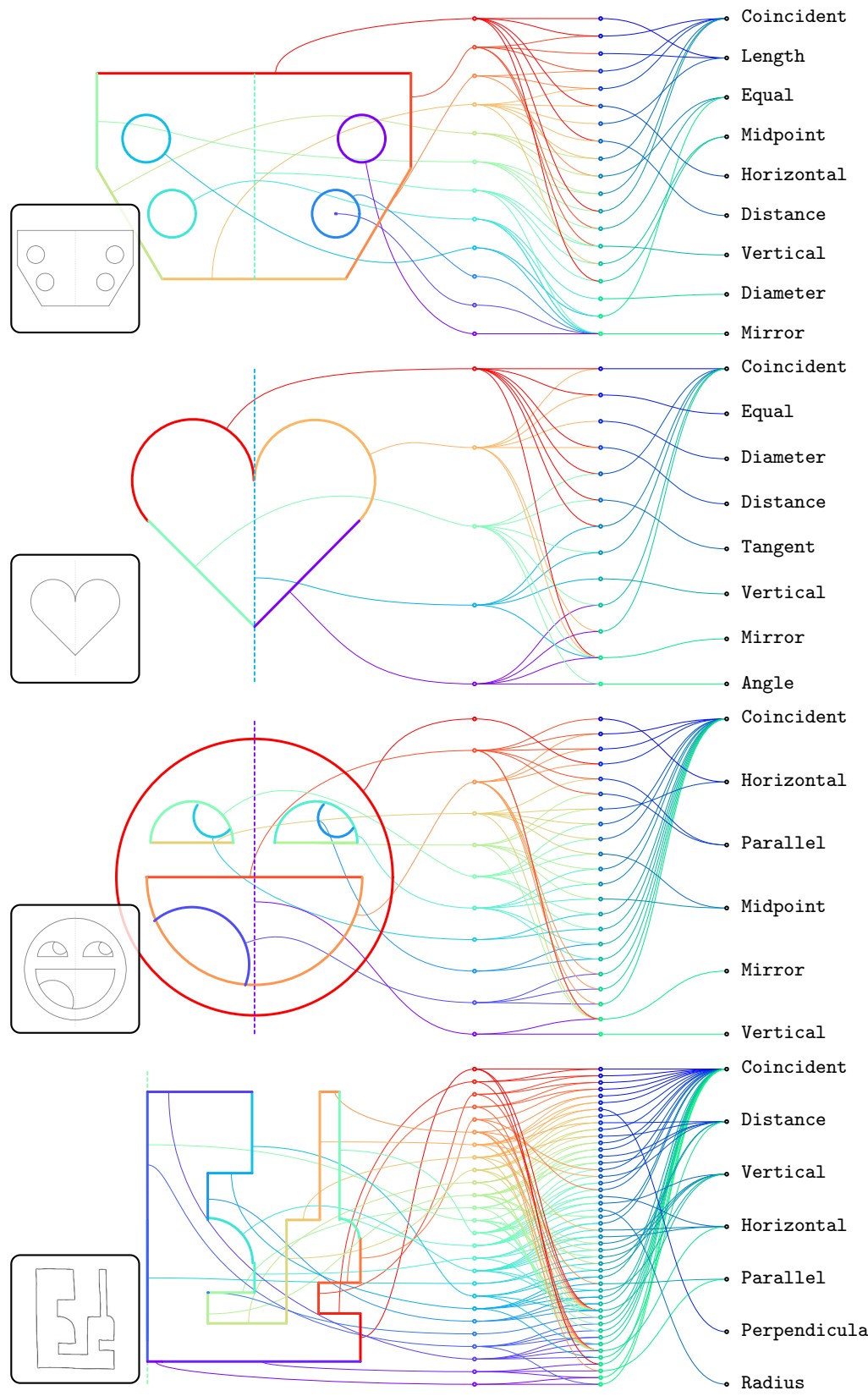

Figure 7: Additional **entities and constraints** sampled the conditional model.

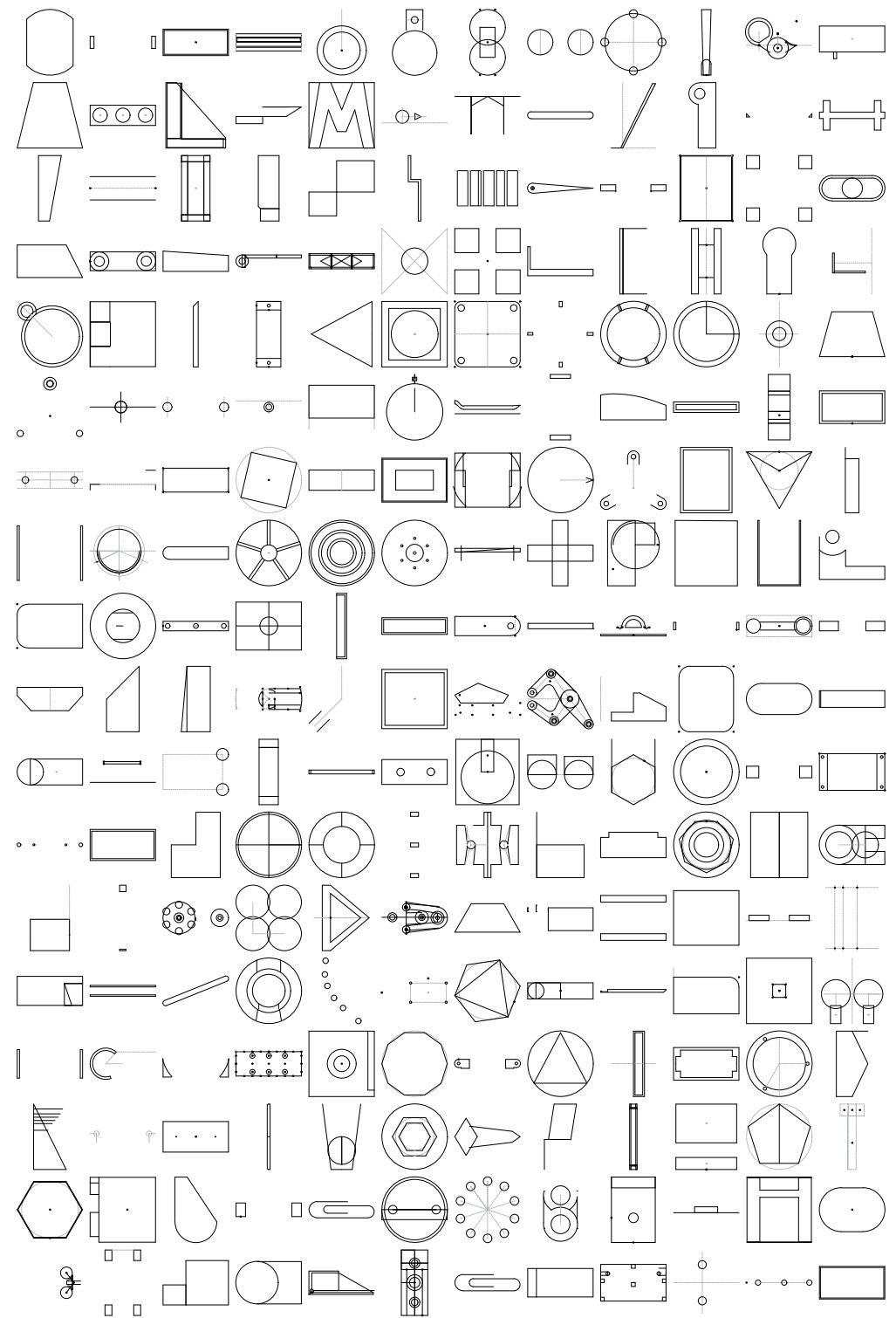

Figure 8: **Additional random sketches** sampled from the unconditional triplet model. We use Nucleus Sampling with *top-p* = 0.9.

# K    A note on efficiency

We note that speed and efficiency were not the focus of this work. Our implementation of the model is a mere proof of concept. A typical training run of $10^6$ steps for a 24-layer Transformer with the hidden dimension of 384 operating on 256 element batches takes about 6 (for a 1024-token triplet model) or 7 (for a 1990-token byte model) days to finish. The triplet model is less efficient since it needs to compute outputs for all possible token groups at every time step. This could be avoided if the hardware allowed for dynamic scattering and gathering (*i.e.*, we could gather tokens of the same group and process them independently from the rest). Unfortunately, TPUs that we use for training do not support this feature.

Sampling from the model is not real time. A batch of 8 sketches (up to 1024 tokens) is generated in $\approx 1$ minute on a TPU v2. Note that large autoregressive Transformers are generally not very fast. Our triplet setting has an additional overhead due to necessary transfers between the device and the host memory at every sampling step as the interpreter resides on the host CPU. Moreover, the interpreter is implemented in pure `python` and not optimized at all. Rewriting it in a more speed-centric fashion (*e.g.*, as a `C++` module) might improve the throughput.

We would also like to point out that we did very little hyperparameter exploration and it is possible that smaller versions of the model can achieve comparable quality while being faster.

# L    Potential improvements to the architecture

When it comes to model architecture, there are a few ways in which we can potentially improve it. For example, we can enhance the input fed into the Transformer at every time step by incorporating a visual representation of a partially constructed sketch. There is evidence that this may lead to better performance [14] as the model gains more direct access to the holistic state of generation. Another improvement we briefly mentioned in Section 5 is tighter interaction between the model and the CAD software which makes ultimate use of the synthesized data. For instance, currently, our training procedure does not take into account how the sketch solver would react to adding more entities and constraints. The model is simply trying to imitate valid examples from the dataset. As a result, the model's capacity is not always utilized efficiently: a lot of it may be allocated on capturing the minutiae of sketch parameters with more important properties like solvability and stability being overlooked. Therefore it makes sense to introduce training objectives that are directly correlated with the desired behaviour of the system. This brings us to the territory of reinforcement learning with rich feedback from the software environment, a setting which is much closer to how sketching is done by human designers.