# OpenReview forum: "Computer-Aided Design as Language"
_NeurIPS.cc/2021/Conference — NeurIPS 2021 Poster_

### Official Review · Reviewer_p1Hz · 2021-07-15

**Rating:** 7
**Confidence:** 4

**Summary:**

This paper proposes a deep learning model of automatically generating computer-aided design (CAD) sketches, which are typically composed of entities (e.g. line and arc) and constraints (e.g. tangent and mirror). The authors devise a method to describe CAD sketches as serialized Protocol Buffers, and further represent them as sequences of triplet tokens. Then they combine the devised data representation with language modeling techniques (Transformer + PointerNet) to capture the data distribution in an auto-regressive manner. The proposed method is evaluated mainly on unconditional generation and various design choices are ablated. Additional generation results are shown conditioned on binary images.

**Limitations And Societal Impact:**

Suggestions are as above.

**Main Review:**

The novelty of this work is clear to me. As one of very first attempts on generative modeling for engineering CAD sketches (there are some concurrent works), it devise a method to represent CAD sketch as a sequence of triplet tokens and combine this representation with general language modeling technique to capture both the geometric primitives and constraints. The proposed special embedding and pointer strategy for its triplet representation is also interesting.

The results in general look good and I like the colored visualization. Besides unconditional generation, one of the potential application, image-conditional generation is also thoroughly evaluated.

My concerns and suggestions are as follows:
1. It does not compare to any prior works. I understand it's the very first work for CAD sketch generation, but there are prior works (e.g. DeepSVG) that generates vector graphics, which are also parametric sketches but without constraints. The authors can just compare results in the image domain. The same applies to the image-conditioned generation.
2. The test set is too small. A test set of 50K is only 1% compared to a training set of 4.6M (line 239-241). This could result in much variance. 90%-10%-10% is the general practice. I hope the author to use a larger test set in the revision. If not, it's better to explain why.
3. Consider to cite concurrent works on CAD generation:
    - DeepCAD: A Deep Generative Network for Computer-Aided Design Models
    - SketchGen: Generating Constrained CAD Sketches
4. What exactly is external sketch solver (line 288)? I guess it's some sort of constraint optimizer. It's better to give some detailed descriptions (including in the supplement is OK).

Lastly, will the dataset be released? A large dataset is always beneficial to the community.


**Time Spent Reviewing:**

5 hours

---

> ### Author Response · Authors · 2021-08-10
> **Response**
>
> We thank the reviewer for the valuable feedback! Please find our responses to the questions / concerns below:
>
> > **Q1** It does not compare to any prior works. I understand it's the very first work for CAD sketch generation, but there are prior works (e.g. DeepSVG) that generates vector graphics, which are also parametric sketches but without constraints. The authors can just compare results in the image domain. The same applies to the image-conditioned generation.
>
> Since in this work, we are considering engineering CAD sketches it would make sense to compare against other methods solving the same problem. Unfortunately, as the reviewer points out, there are no established baselines in this area. The only work that can be deemed prior to ours (although we developed our method concurrently and independently and did not employ any of their ideas / data) is SketchGraphs *[1]*. Comparing to SketchGraphs is difficult due to differences in the data. For example, their system supports only a subset of the DSL that we use in our work. Moreover, the generative model presented in *[1]* does not predict parameters of the sketch entities – this makes likelihood comparison problematic (also mentioned in *[2]*). We made an attempt to convert our dataset into the format suitable for *[1]* and after substantial filtering managed to launch the original training code. The resulting samples look very similar to the last column of Figure 5 in *[2]*, i.e., a brief visual inspection is enough to conclude that the model produces sketches that are far from the real data. We will include this finding in the final version of our manuscript.
>
> Regarding generative models for vector graphics (e.g., DeepSVG *[4]*), we believe that comparing against such systems would not be fair. It is, indeed, possible to drop the constraint information from CAD sketches and treat our method as a generative model for plain vector images. This, however, puts our approach at disadvantage. The goal of our paper was to demonstrate how objects of arbitrary structures, be that entities or constraints, can be generated in a unified fashion using the same model. While we focused on designing our method to be flexible and domain-agnostic, we largely disregarded the data design (we closely follow the native Onshape format which may not be the best for vector images) and domain-specific architectural choices / tricks  (e.g., DeepSVG takes advantage of the fact that the next primitive starts where the previous one ends – we don’t). It’s possible that a system tailored to SVGs will outperform our approach at generating SVGs.
>
> Lastly, we would like to point out that constraints are an important aspect of CAD sketches since (besides conveying the design intent) they can fix some of the mistakes in the prediction of entity parameters (as shown in Figure 3d). We compared the samples produced by our model before and after applying the sketch solver (i.e., enforcing the constraints) and found that the distribution of the number of closed regions in the latter case is closer to the real data (in terms of 1D Wasserstein distance to the data distribution):
> ```
> Model         | w/ constraints | w/o constraints |
> --------------------------------------------------
> Byte (0.9)    | 3.24e-3        | 4.69e-3         |
> Byte (1)      | 2.94e-3        | 3.68e-3         |
> Triplet (0.9) | 1.30e-3        | 1.99e-3         |
> Triplet (1)   | 1.94e-3        | 2.74e-3         |
> ```
>
> > **Q2** The test set is too small. A test set of 50K is only 1% compared to a training set of 4.6M (line 239-241). This could result in much variance. 90%-10%-10% is the general practice. I hope the author to use a larger test set in the revision. If not, it's better to explain why.
>
> We took the test size directly from *[1]*. Concurrent works like *[2, 3]* also don’t seem to follow the 80%-10%-10% practice.
>
> > **Q3** Consider to cite concurrent works on CAD generation:
> > * DeepCAD: A Deep Generative Network for Computer-Aided Design Models
> > * SketchGen: Generating Constrained CAD Sketches
>
> We thank the reviewer for bringing this relevant concurrent research to our attention! We will cite these papers in the final version of the manuscript.
>
> > **Q4** What exactly is external sketch solver (line 288)? I guess it's some sort of constraint optimizer. It's better to give some detailed descriptions (including in the supplement is OK).
>
> Yes, the external sketch solver is an algorithm for enforcing the constraints. The parameters of the entities produced by our model are “initial guesses” which are adjusted by the optimization procedure until all the constraints are satisfied. We will include this clarification in the camera-ready version of the paper.
>
> > **Q5** Lastly, will the dataset be released? A large dataset is always beneficial to the community.
>
> Yes, we will release the dataset. In fact, it’s going through a legal review at the moment and should become publicly available in the next few weeks.
>
> ### References
> * *[1]* Seff et al., "SketchGraphs: A Large-Scale Dataset for Modeling Relational Geometry in Computer-Aided Design", 2020
> * *[2]* Willis et al., "Engineering Sketch Generation for Computer-Aided Design", 2021
> * *[3]* Para et al., "SketchGen: Generating Constrained CAD Sketches", 2021
> * *[4]* Carlier et al., "DeepSVG: A Hierarchical Generative Network for Vector Graphics Animation", 2020

---

### Official Review · Reviewer_K3kn · 2021-07-19

**Rating:** 6
**Confidence:** 4

**Summary:**

This paper provides a large dataset of structured 2D sketches and proposes transformer-based models capable of both unconditional and conditional generations of such sketches. The authors present a novel representation for structured sketches based on Protocol Buffers, encoding both objects and the constraint between them, and further devise a custom tokenization scheme to adapt it for the training and inference using a transformer decoder architecture, showing advantages compared to a simple byte tokenization. Their model can successfully generate valid structured sketches from scratch, and moreover infer valid objects and constraints from an input image (as a condition).

**Limitations And Societal Impact:**

Ok

**Main Review:**

The paper is mostly well-written and easy to follow. The proposed coding of structured 2D sketches is novel and seems useful, particularly for an effective correction of predicted objects based on constraints. The dataset, which the authors plan to make public, could be useful to other applications due to its quality and structured labeling, and the experiments, though not extensive, show that their proposed models work as expected. I have a few concerns and questions that I will elaborate below:

1. Run-time and train-time of the models are missing, and this makes comparisons between models and an overall efficiency/practicality evaluation hard. For example, how far is the generation (unconditional and conditional) from real-time performance?

2.  Comparison to similar models is missing, that is, how does the model compare in quality and speed with other vector image generation methods (conditional and unconditional)? (an extension of table 4 in appendix H).

3. Whenever reporting average, it is good practice to report standard deviation as well, to provide a sense of confidence.

4. It is important to highlight/discuss the types of failures (wrong shapes and wrong constraints) and the rates with which each happen, to give a better picture of the models’ capabilities.  Also, I think discussion of model failure, such as the compression failure discussed in appendix G, are very valuable to the more broad transformer research, and are a good idea to include and elaborate on in the main text.

Side note: wrong syntax in Eq.1.

**Time Spent Reviewing:**

5

---

> ### Author Response · Authors · 2021-08-10
> **Response**
>
> We thank the reviewer for the valuable feedback! Please find our responses to the questions / concerns below:
>
> > **Q1** Run-time and train-time of the models are missing, and this makes comparisons between models and an overall efficiency/practicality evaluation hard. For example, how far is the generation (unconditional and conditional) from real-time performance?
>
> Firstly, we would like to note that speed / efficiency was not the focus of our work. Our implementation of the model is a mere proof of concept and therefore there is definitely plenty of room for improvement in this direction.
>
> That said, a typical training run of 1e6 steps for a 24-layer Transformer with the hidden dimension of 384 operating on 256 element batches takes about 6 (for a 1024-token triplet model) or 7 (for a 1990-token byte model) days to finish. The triplet model is less efficient since it needs to compute outputs for all possible token groups at every timestep. This could be avoided if the hardware allowed for dynamic scattering / gathering (i.e., we could gather tokens of the same group and process them independently from the rest). Unfortunately, TPUs that we use for training don’t support this feature.
>
> Sampling from the model is not real time. A batch of 8 sketches (up to 1024 tokens) is generated in ~1 minute on a TPU v2. Note that large autoregressive Transformers are generally not very fast. Our triplet setting has an additional overhead due to necessary transfers between the device and host memory at every sampling step (the interpreter resides on the host CPU). Moreover, the interpreter is implemented in pure python and not optimized at all. Rewriting it in a more speed-centric fashion (say, as a C++ module) might improve the throughput.
>
> We would also like to point out that we did very little hyperparameter exploration and it’s possible that smaller versions of the model can achieve comparable quality while being faster.
>
> > **Q2** Comparison to similar models is missing, that is, how does the model compare in quality and speed with other vector image generation methods (conditional and unconditional)? (an extension of table 4 in appendix H).
>
> Inference speed is indeed important for large-scale adoption of machine learning models, however, as we explain above, it was not the focus of our study and our implementation was solely optimized for the ease of experimentation. For the discussion on the quality comparison we refer the reviewer to our responses to **Q2** of **Reviewer KVBt** and **Q1** of **Reviewer p1Hz**.
>
> > **Q3** Whenever reporting average, it is good practice to report standard deviation as well, to provide a sense of confidence.
>
> We decided to omit standard deviations since obtaining results for multiple seeds is a costly procedure (one experiment may take up to a week to finish). Moreover, we didn’t observe any significant variation in test scores between training runs during development.
>
> > **Q4** It is important to highlight/discuss the types of failures (wrong shapes and wrong constraints) and the rates with which each happen, to give a better picture of the models’ capabilities. Also, I think discussion of model failure, such as the compression failure discussed in appendix G, are very valuable to the more broad transformer research, and are a good idea to include and elaborate on in the main text.
>
> First, we would like to point out that we report proportions of invalid (unsolvable or malformed) sketches produced by our model in lines 270-275 of the manuscript.
>
> When it comes to more semantic types of failures, we note that our system at its core is a domain-agnostic Transformer-based language model operating on sequences of discrete (or quantized) values and therefore inherits typical problems of such models. For example, imperfect autoregressive LMs suffer from not being able to correct their own mistakes made earlier in the sequence (mentioned in lines 279-283 in the manuscript). This behaviour manifests itself in occasional samples cluttered with “garbage” entities. Another tendency of the model is generation of simpler (“boring”) sketches – this stems from the fact that we are relying on a repository of CAD shapes created in a “free-to-use” software and therefore often authored by inexperienced users. Lastly, we observe a significant number of synthesized shapes that don’t form closed regions due to the lack of precision in prediction of entity parameters. This can be partially explained by how we treat floating-point fields – we describe quantized parameters using the categorical distribution which doesn’t assume any ordering of the values. The latter is done to simplify training but, in principle, our framework is not restricted to discrete nominal tokens. We leave this direction for future work.

---

> > ### Comment · Reviewer_K3kn · 2021-08-28
> > **Final thoughts**
> >
> > Thank you for your answers. Including these discussions, particularly on computational time and practicality, in your final manuscript will be very helpful.
> >
> > After reading all the reviews and the responses, in my opinion, the paper, while an interesting proof-of-concept, still lacks the extensive experimentation that makes proof-of-concept models valuable (i.e. a good guide for future improvements and practical adaptations), so I keep my current score at borderline.

---

> > > ### Author Response · Authors · 2021-08-31
> > > **Thank you for your feedback**
> > >
> > > Thank you for your feedback! We will include the discussion above into our final version.
> > >
> > > We would like to point out that our paper does contain quantitative (at least 3 different metrics and a new benchmark dataset) and qualitative evaluation useful for future improvements. In our rebuttal, we explained why it's difficult to obtain other quality metrics and why comparing against more baselines is not a trivial task (in short, there are no established benchmarks and baselines in this area). We would greatly appreciate if you could provide more specific suggestions for experimentation that would significantly improve the quality of the paper.

---

### Official Review · Reviewer_KVBt · 2021-07-26

**Rating:** 6
**Confidence:** 4

**Summary:**

This paper proposes a novel method to represent structured CAD sketch objects via Protocol Buffers
and demonstrate its generalization ability across the heterogeneous nature of sketches. They
also proposed a transformer-based generative model for generating the CAD sketches and evaluate
its effectiveness on carefully collected and preprocessed data which consists of over 4.7M
sketches. They show the capability of the generative model for both unconditional and conditional
generation tasks.

**Ethical Concerns:**

No.

**Limitations And Societal Impact:**

This paper shows strong motivation for automated CAD sketches generation. The proposed
sequential data representation takes the advantage of using the transformer-based architecture
for the sketch generation. Yet, this paper lacks some important evaluation metrics and
baseline comparisons, which is the main reason for my rating not being higher.
The experiment results show the high generation quality of the CAD
sketches generation for both conditional, and unconditional tasks.

**Main Review:**

*Strengths*:
1. In general, the motivation of the paper is quite strong. There are few learning-based methods
for generating CAD sketches.
2. The proposed data representation costs less space than the classic CAD sketch format by
removing unnecessary information.
3. The proposed triplet and byte representation overcome the necessity of converting the raw
sketch message into a text format that contains both the structure and content of the data.
4. The quantitative (Table 2) and qualitative (Figure 3) experiment results demonstrate the better
performance of the triple representations than byte ones.
5. The predicted constraints by the model are reasonable for making a closed-shape CAD sketch.
6. The proposed representation methods lead to a high generation quality for both conditional
and unconditional generation tasks.

*Weakness*:
1. The metric for evaluating the performance of the proposed method is not sufficient. How about the
fidelity and diversity of the generation with respect to the training dataset? Those should be quantitatively evaluated also.
2. The paper lack comparison with some existing generative models for sketch generation, such as Sketch-RNN.
3. I think it is necessary to compare the proposed representation with the naïve text format.
4. Line 141 has a typo. No Figure 1b.

**Time Spent Reviewing:**

2 hours

---

> ### Author Response · Authors · 2021-08-10
> **Response**
>
> We thank the reviewer for the valuable feedback! Please find our responses to the questions / concerns below:
>
> > **Q1** The metric for evaluating the performance of the proposed method is not sufficient. How about the fidelity and diversity of the generation with respect to the training dataset? Those should be quantitatively evaluated also.
>
> We would like to point out that besides the test likelihoods (Table 2) we also provide several other performance metrics. For example, Figure 4 along with Figure 6 (in the Appendix) contain a variety of sketch statistics computed both for the samples and the data. According to these statistics the sketches synthesized by our models are reasonably close to the held-out real data. For the image-to-sketch setting, we provide a comparison against the nearest neighbour baseline using Chamfer distance as reconstruction metric (Table 4 in the Appendix).
>
> The majority of the off-the-shelf fidelity / diversity metrics for image-generative models rely on some semantic representation computed by a dedicated neural network. It’s a non-trivial task to define such a representation for the CAD sketch data – for example, features extracted by a network trained on ImageNet (the most common approach) won’t be appropriate here since ImageNet is very different in appearance from the CAD line art. That said, if the reviewer has a concrete suggestion for the metric they think would be a good fit for the domain we would gladly provide it in the camera-ready version of the paper.
>
> > **Q2** The paper lack comparison with some existing generative models for sketch generation, such as Sketch-RNN.
>
> We would like to point out that the setting considered in the Sketch-RNN paper *[1]* is significantly different from ours. There they only deal with sequences of straight line segments and this representation would not be sufficient to describe CAD sketches. For further discussion please refer to our response to **Q1** of **Reviewer p1Hz**.
>
> > **Q3** I think it is necessary to compare the proposed representation with the naïve text format.
>
> The difficulty with the text format is that sequences can get very long depending on the chosen tokenization strategy. For example, the SentencePiece *[2]* tokenizer that splits tokens at whitespaces yields training examples that can be as long as 6100 tokens which exceeds our computational budget. After playing around with the settings we arrived at a reasonably sized tokenized representation with the vocabulary of 8000 words. Training a plain Transformer-based language model yields the following test likelihoods: `4.655 bits per object, 141.055 bits per sketch`. This result sits in between the random baselines and the byte model trained on the concatenated sequences (Table 2).
>
> > **Q4** Line 141 has a typo. No Figure 1b.
>
> We thank the reviewer for spotting this one! We will correct the typo in the camera-ready version.
>
> ### References
> * *[1]* Ha & Eck, "A Neural Representation of Sketch Drawings", 2017
> * *[2]* Kudo & Richardson, "SentencePiece: A simple and language independent subword tokenizer and detokenizer for Neural Text Processing", 2018

---

### Official Review · Reviewer_yGks · 2021-07-27

**Rating:** 7
**Confidence:** 3

**Summary:**

This paper presents a learning-based model to generate 2D sketches that are widely used in 3D CAD applications. It considers 2D sketches consisting of entities (lines, curves, etc.) and constraints (mirror, coincide, perpendicular, etc.), a representation that is commonly used in modern CAD software. By design, 2D sketches are structured objects. To handle these structured data, this paper presents a linear representation using Protocol Buffers, which flattens the structured data into a sequence of triplets. Through this representation, the paper establishes similarities between natural language modeling and 2D sketches and proposes to use Transformer to train a generative model that captures the distribution of sketches in a large dataset. The paper reports quantitative and qualitative performance of the generative model.

**Limitations And Societal Impact:**

Please see my review above for limitations. I do not see any major issues with negative societal impact from this paper.

**Main Review:**

This paper is dealing with a practical problem that has wide industrial applications. The high-level idea behind the proposed technical method (bridging 2D sketches and languages) makes sense to me. The quantitative and qualitative evaluation seems OK, although I think a few more experiments could be added to make the paper stronger (see below). Overall, I think this paper has contributed some technical ideas that this community can benefit from. However, I still have a few questions and would like to make my final decision after reading the authors’ answers:
- A high-level comment on the motivation and application of the technical method: the paper attempts to make an analogy between 2D sketches and natural language modeling. They are indeed similar in some way but also have differences. In particular, 2D sketches in mechanical designs have a low tolerance of inaccurate or imprecise designs, while imprecise natural languages are quite acceptable in many applications. As an example, 2D sketches of a pair of a nut and a bolt must have compatible radii, but a generative model like this paper might generate designs with roughly the same radii which, while looking similar, won’t work at all in practice. To what extent can we extend this Transformer idea to generate precise 2D sketches that are really usable in real-world mechanical design applications?
- The paper mentioned that long sequences of data are a challenge, motivating the usage of PB messages. From what I understand, the proposed data representation still duplicates geometric information, e.g., line 4-5 and line 11-12 in Table 1 store the same point (x, y) twice. Why not separate topology and geometry information like in a boundary representation (B-rep), or did I misunderstand your data representation? What is fundamentally novel in your Protocol Buffer representation if I compare it with B-rep?
- The paper mentioned “from coffee mugs to sports cars” twice in the main paper and its supplemental material, yet the 2D sketches shown in the paper are not even close to the complexity of a sports car. In particular, line 513-515 seems to imply some of your sketches are from sports cars. If you have impressive results from these sophisticated models, please add them to the main paper, otherwise, I would suggest toning down this description and state what examples your 2D sketches can handle in a more factual way.
- One thing that makes 2D sketches really useful in CAD applications is that they are parametric designs: the (entity, constraint) combo not only defines a particular design itself but also provides a family of similar designs if you perturb the continuous parameters (e.g., length of a line, angle of an arc) in these entities. In practice, it is surprisingly not uncommon for these perturbations to result in invalid sketches that break some constraints if constraints are not defined properly. I would love to see some small experiments on the 2D sketches generated by this paper, i.e., mildly perturbing the entity parameters in the resultant 2D sketches, and see how often they lead to invalid sketches that break the constraints. I understand this may be out of the scope of this paper, but I think having such an experiment would be helpful for us to fully understand how well these generated 2D sketches can be used for parametric designs.


**Time Spent Reviewing:**

4

---

> ### Author Response · Authors · 2021-08-10
> **Response**
>
> We thank the reviewer for the valuable feedback! Please find our responses to the questions / concerns below:
>
> > **Q1** A high-level comment on the motivation and application of the technical method: the paper attempts to make an analogy between 2D sketches and natural language modeling. They are indeed similar in some way but also have differences. In particular, 2D sketches in mechanical designs have a low tolerance of inaccurate or imprecise designs, while imprecise natural languages are quite acceptable in many applications. As an example, 2D sketches of a pair of a nut and a bolt must have compatible radii, but a generative model like this paper might generate designs with roughly the same radii which, while looking similar, won’t work at all in practice. To what extent can we extend this Transformer idea to generate precise 2D sketches that are really usable in real-world mechanical design applications?
>
> First, we would like to note that language models can be made quite accurate given enough data and compute. The appeal of LMs lies in the fact that they require relatively little domain knowledge to work reasonably well. This appeal becomes especially pronounced in less explored (by the ML community) areas such as CAD. With our paper we wanted to demonstrate that a domain-agnostic LM-based system is capable of achieving good results in a new domain of CAD sketches and through this we hoped to make the case of using LMs stronger.
>
> We agree that a naive approach to representing the data as language may sometimes yield suboptimal results like in the example mentioned by the reviewer. This, however, can be at least partially addressed by using more domain knowledge whilst designing the DSL. For example, our model may make a mistake by predicting non-matching centers of two circles which are intended to be concentric. Luckily, this lack of precision is not fatal since the model can additionally emit a “concentric” constraint that will force two centers to coincide. Thus, the model does not need to be very precise to still produce valid CAD sketches. In the reviewer’s example, a similar technique could be employed to make the radii compatible. We believe that injecting domain knowledge into the language is a promising approach enjoying several advantages over traditional reliance on domain-specific model architectures (e.g., it’s less restrictive and may be employed by non-experts in ML).
>
> > **Q2** The paper mentioned that long sequences of data are a challenge, motivating the usage of PB messages. From what I understand, the proposed data representation still duplicates geometric information, e.g., line 4-5 and line 11-12 in Table 1 store the same point (x, y) twice. Why not separate topology and geometry information like in a boundary representation (B-rep), or did I misunderstand your data representation? What is fundamentally novel in your Protocol Buffer representation if I compare it with B-rep?
>
> In our experiments, we closely followed the original Onshape format – we simply translated it to PB messages with only minor changes borrowing the structures of objects from the original JSONs. We tried to stay as domain-agnostic as possible so we didn’t spend much time on data design. As a result, our system should be applicable to a variety of use cases beyond CAD sketch generation. The savings in sequence lengths are coming “for free” just because we employ a more efficient off-the-shelf serialization strategy (PB vs JSON).
>
> That said, the “B-Rep way” of structuring the data will likely boost the performance of the model albeit at the cost of a more involved data massaging. It is possible to rewrite the definitions of entities to separate topology and geometry information and our framework should be able to handle this format since we already support pointers to other objects in the definitions of constraints. We think it’s a promising direction for future work!
>
> > **Q3** The paper mentioned “from coffee mugs to sports cars” twice in the main paper and its supplemental material, yet the 2D sketches shown in the paper are not even close to the complexity of a sports car. In particular, line 513-515 seems to imply some of your sketches are from sports cars. If you have impressive results from these sophisticated models, please add them to the main paper, otherwise, I would suggest toning down this description and state what examples your 2D sketches can handle in a more factual way.
>
> The references to coffee mugs and sports cars were made solely to explain to a reader currently not familiar with CAD the breadth of the spectrum of objects designed using this paradigm. We, of course, don’t claim that our model can generate a sports car, merely because very few (if any) full sketches of sports cars appear in our training data, but would not immediately reject the possibility of our model being able to generate a small part of a sports car.
>
> > **Q4** One thing that makes 2D sketches really useful in CAD applications is that they are parametric designs: the (entity, constraint) combo not only defines a particular design itself but also provides a family of similar designs if you perturb the continuous parameters (e.g., length of a line, angle of an arc) in these entities. In practice, it is surprisingly not uncommon for these perturbations to result in invalid sketches that break some constraints if constraints are not defined properly. I would love to see some small experiments on the 2D sketches generated by this paper, i.e., mildly perturbing the entity parameters in the resultant 2D sketches, and see how often they lead to invalid sketches that break the constraints. I understand this may be out of the scope of this paper, but I think having such an experiment would be helpful for us to fully understand how well these generated 2D sketches can be used for parametric designs.
>
> This experiment should be fairly easy to conduct and we can include it in the paper but we are not sure we completely understand the ultimate goal. Are we only interested in whether or not the perturbed sketch is still solvable or do we want to see if the design intent is preserved under transformations? In the latter case, even if a perturbation leads to a solvable sketch the solution may not have the original design intent.

---

> > ### Comment · Reviewer_yGks · 2021-08-20
> > **Thank you for your response**
> >
> > Thank you very much for your detailed reply.
> >
> > I suggest you consider merging your answer to Q2 on B-rep vs. Onshape representation to your main paper (e.g., in intro or related work).
> >
> > The last experiment is not a deal-breaker to me. I don't have further questions.
> >
> > Looking forward to reading your revised manuscript!

---

> > > ### Author Response · Authors · 2021-08-31
> > > **Thank you**
> > >
> > > Thank you for your response! We will incorporate our answer to Q2 into our manuscript.

---

### Decision · Program_Chairs · 2021-09-27

**Decision:**

Accept (Poster)

**Comment:**

This paper presents a new approach to generating constrained CAD sketches. The key challenge in this problem is generating constraints that relate all the different strokes in a sketch. At a high level, the main idea in this paper is to treat this as a language generation problem, since the sketches with their constraints can be represented in a language. The key new idea for this paper is that instead of learning directly in the language of CAD sketches, they first encode the strings to a very compact and dense representation, which allows the network to learn more efficiently. Versions of this idea have been used in other settings but the idea is new in this context.

Additionally, the paper introduces a new dataset for this problem which should be of significant value to the community.